# Self-Paced Learning with Diversity

**Lu Jiang[1], Deyu Meng[1,2], Shoou-I Yu[1], Zhenzhong Lan[1], Shiguang Shan[1,3], Alexander G. Hauptmann[1]**
[1]School of Computer Science, Carnegie Mellon University
[2]School of Mathematics and Statistics, Xi'an Jiaotong University
[3]Institute of Computing Technology, Chinese Academy of Sciences
`lujiang@cs.cmu.edu, dymeng@mail.xjtu.edu.cn`
`{iyu, lanzhzh}@cs.cmu.edu, sgshan@ict.ac.cn, alex@cs.cmu.edu`

## Abstract

Self-paced learning (SPL) is a recently proposed learning regime inspired by the learning process of humans and animals that gradually incorporates easy to more complex samples into training. Existing methods are limited in that they ignore an important aspect in learning: diversity. To incorporate this information, we propose an approach called self-paced learning with diversity (SPLD) which formalizes the preference for both easy and diverse samples into a general regularizer. This regularization term is independent of the learning objective, and thus can be easily generalized into various learning tasks. Albeit non-convex, the optimization of the variables included in this SPLD regularization term for sample selection can be globally solved in linearithmic time. We demonstrate that our method significantly outperforms the conventional SPL on three real-world datasets. Specifically, SPLD achieves the best MAP so far reported in literature on the Hollywood2 and Olympic Sports datasets.

## 1 Introduction

Since it was raised in 2009, *Curriculum Learning* (CL) [1] has been attracting increasing attention in the field of machine learning and computer vision [2]. The learning paradigm is inspired by the learning principle underlying the cognitive process of humans and animals, which generally starts with learning easier aspects of an aimed task, and then gradually takes more complex examples into consideration. It has been empirically demonstrated to be beneficial in avoiding bad local minima and in achieving a better generalization result [1].

A sequence of gradually added training samples [1] is called a curriculum. A straightforward way to design a curriculum is to select samples based on certain heuristical "easiness" measurements [3, 4, 5]. This ad-hoc implementation, however, is problem-specific and lacks generalization capacity. To alleviate this deficiency, Kumar et al. [6] proposed a method called *Self-Paced Learning* (SPL) that embeds curriculum designing into model learning. SPL introduces a regularization term into the learning objective so that the model is jointly learned with a curriculum consisting of easy to complex samples. As its name suggests, the curriculum is gradually determined by the model itself based on what it has already learned, as opposed to some predefined heuristic criteria. Since the curriculum in the SPL is independent of model objectives in specific problems, SPL represents a general implementation [7, 8] for curriculum learning.

In SPL, samples in a curriculum are selected solely in terms of "easiness". In this work, we reveal that diversity, an important aspect in learning, should also be considered. Ideal self-paced learning should utilize not only easy but also diverse examples that are sufficiently dissimilar from what has already been learned. Theoretically, considering diversity in learning is consistent with the increasing entropy theory in CL that a curriculum should increase the diversity of training examples [1]. This can be intuitively explained in the context of human education. A rational curriculum for a pupil not only needs to include examples of suitable easiness matching her learning pace, but also,

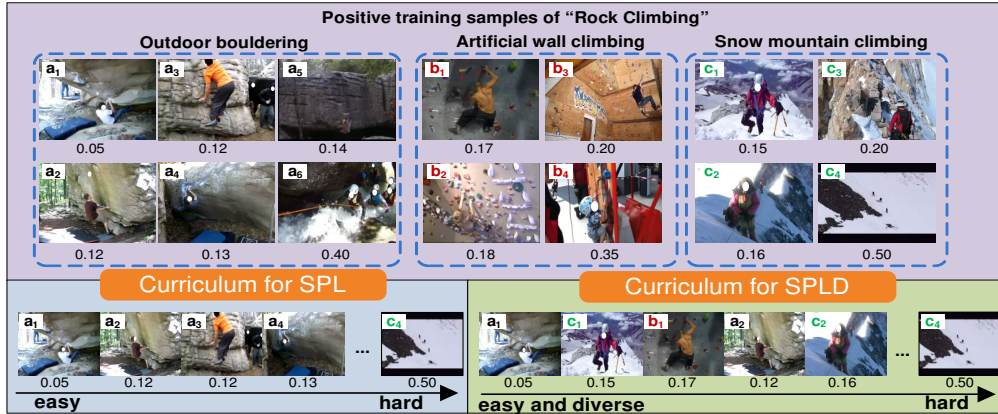

Figure 1: Illustrative comparison of SPL and SPLD on "Rock Climbing" event using real samples [15]. SPL tends to first select the easiest samples from a single group. SPLD inclines to select easy and diverse samples from multiple groups.

importantly, should include some diverse examples on the subject in order for her to develop more comprehensive knowledge. Likewise, learning from easy and diverse samples is expected to be better than learning from either criterion alone.

We name the learning paradigm that considers both easiness and diversity *Self-Paced Learning with Diversity* (SPLD). SPLD proves to be a general learning framework as its intuition is embedded as a regularization term that is independent of specific model objectives. In addition, by considering diversity in learning, SPLD is capable of obtaining better solutions. For example, Fig. 1 plots some positive samples for the event "Rock Climbing" on a real dataset, named MED [15]. Three groups of samples are depicted for illustration. The number under the keyframe indicates the loss, and a smaller loss corresponds to an easier sample. Every group has easy and complex samples. Having learned some samples from a group, the SPL model prefers to select more samples from the same group as they appear to be to what the model has learned. This may lead to overfitting to a data subset while ignoring easy samples in other groups. For example, in Fig. 1, the samples selected in first iterations of SPL are all from the "Outdoor bouldering" sub-event because they all look like $a_1$. This is significant as the overfitting becomes more and more severe as the samples from the same group are kept adding into training. This phenomenon is more evident in real-world data where the collected samples are usually biased towards some groups. In contrast, SPLD, considering both easiness and diversity, produces a curriculum that reasonably mixes easy samples from multiple groups. The diverse curriculum is expected to help quickly grasp easy and comprehensive knowledge and to obtain better solutions. This hypothesis is substantiated by our experiments.

The contribution of this paper is threefold: (1) We propose a novel idea of considering both easiness and diversity in the self-paced learning, and formulate it into a concise regularization term that can be generally applied to various problems (Section 4.1). (2) We introduce the algorithm that globally optimizes a non-convex problem w.r.t. the variables included in this SPLD regularization term for sample selection (Section 4.2). (3) We demonstrate that the proposed SPLD significantly outperforms SPL on three real-word datasets. Notably, SPLD achieves the best MAP so far reported in literature on two action datasets.

## 2 Related work

Bengio et al. [1] proposed a new learning paradigm called *curriculum learning (CL)*, in which a model is learned by gradually including samples into training from easy to complex so as to increase the entropy of training samples. Afterwards, Bengio and his colleagues [2] presented insightful explorations for the rationality underlying this learning paradigm, and discussed the relationship between the CL and conventional optimization techniques, e.g., the continuation and annealing methods. From human behavioral perspective, Khan et al. [10] provided evidence that CL is consistent with the principle in teaching. The curriculum is often derived by predetermined heuristics in particular problems. For example, Ruvolo and Eaton [3] took the negative distance to the boundary as the indicator for easiness in classification. Spitkovsky et al. [4] used the sentence length as an indicator in

studying grammar induction. Shorter sentences have fewer possible solutions and thus were learned earlier. Lapedriza et al. [5] proposed a similar approach by first ranking examples based on certain "training values" and then greedily training the model on these sorted examples.

The ad-hoc curriculum design in CL turns out onerous or conceptually difficult to implement in different problems. To alleviate this issue, Kumar et al. [6] designed a new formulation, called *self-paced learning (SPL)*. SPL embeds curriculum design (from easy to more complex samples) into model learning. By virtue of its generality, various applications based on the SPL have been proposed very recently [7, 8, 11, 12, 13]. For example, Jiang et al. [7] discovered that pseudo relevance feedback is a type of self-paced learning which explains the rationale of this iterative algorithm starting from the easy examples i.e. the top ranked documents/videos. Tang et al. [8] formulated a self-paced domain adaptation approach by training target domain knowledge starting with easy samples in the source domain. Kumar et al. [11] developed an SPL strategy for the specific-class segmentation task. Supančič and Ramanan [12] designed an SPL method for long-term tracking by setting smallest increase in the SVM objective as the loss function. To the best of our knowledge, there has been no studies to incorporate diversity in SPL.

## 3   Self-Paced Learning

Before introducing our approach, we first briefly review the SPL. Given the training dataset $\mathcal{D} = \{(\mathbf{x}_1, y_1), \cdots, (\mathbf{x}_n, y_n)\}$, where $\mathbf{x}_i \in \mathbb{R}^m$ denotes the $i^{th}$ observed sample, and $y_i$ represents its label, let $L(y_i, f(\mathbf{x}_i, \mathbf{w}))$ denote the loss function which calculates the cost between the ground truth label $y_i$ and the estimated label $f(\mathbf{x}_i, \mathbf{w})$. Here $\mathbf{w}$ represents the model parameter inside the decision function $f$. In SPL, the goal is to jointly learn the model parameter $\mathbf{w}$ and the latent weight variable $\mathbf{v} = [v_1, \cdots, v_n]$ by minimizing:

$$\min_{\mathbf{w}, \mathbf{v}} \mathbb{E}(\mathbf{w}, \mathbf{v}; \lambda) = \sum_{i=1}^{n} v_i L(y_i, f(\mathbf{x}_i, \mathbf{w})) - \lambda \sum_{i=1}^{n} v_i, \text{ s.t. } \mathbf{v} \in [0, 1]^n, \tag{1}$$

where $\lambda$ is a parameter for controlling the learning pace. Eq. (1) indicates the loss of a sample is discounted by a weight. The objective of SPL is to minimize the weighted training loss together with the negative $l_1$-norm regularizer $-\|\mathbf{v}\|_1 = -\sum_{i=1}^{n} v_i$ (since $v_i \geq 0$). This regularization term is general and applicable to various learning tasks with different loss functions [7, 11, 12].

ACS (Alternative Convex Search) is generally used to solve Eq. (1) [6, 8]. It is an iterative method for biconvex optimization, in which the variables are divided into two disjoint blocks. In each iteration, a block of variables are optimized while keeping the other block fixed. When $\mathbf{v}$ is fixed, the existing off-the-shelf supervised learning methods can be employed to obtain the optimal $\mathbf{w}^*$. With the fixed $\mathbf{w}$, the global optimum $\mathbf{v}^* = [v_1^*, \cdots, v_n^*]$ can be easily calculated by [6]:

$$v_i^* = \begin{cases} 1, & L(y_i, f(\mathbf{x}_i, \mathbf{w})) < \lambda, \\ 0, & \text{otherwise.} \end{cases} \tag{2}$$

There exists an intuitive explanation behind this alternative search strategy: 1) when updating $\mathbf{v}$ with a fixed $\mathbf{w}$, a sample whose loss is smaller than a certain threshold $\lambda$ is taken as an "easy" sample, and will be selected in training ($v_i^* = 1$), or otherwise unselected ($v_i^* = 0$); 2) when updating $\mathbf{w}$ with a fixed $\mathbf{v}$, the classifier is trained only on the selected "easy" samples. The parameter $\lambda$ controls the pace at which the model learns new samples, and physically $\lambda$ corresponds to the "age" of the model. When $\lambda$ is small, only "easy" samples with small losses will be considered. As $\lambda$ grows, more samples with larger losses will be gradually appended to train a more "mature" model.

## 4   Self-Paced Learning with Diversity

In this section we detail the proposed learning paradigm called SPLD. We first formally define its objective in Section 4.1, and discuss an efficient algorithm to solve the problem in Section 4.2.

### 4.1   SPLD Model

Diversity implies that the selected samples should be less similar or clustered. An intuitive approach for realizing this is by selecting samples of different groups scattered in the sample space. We assume that the correlation of samples between groups is less than that of within a group. This

auxiliary group membership is either given, e.g. in object recognition frames from the same video can be regarded from the same group, or can be obtained by clustering samples.

This aim of SPLD can be mathematically described as follows. Assume that the training samples $\mathbf{X} = (\mathbf{x}_1, \cdots, \mathbf{x}_n) \in \mathbb{R}^{m \times n}$ are partitioned into $b$ groups: $\mathbf{X}^{(1)}, \cdots, \mathbf{X}^{(b)}$, where columns of $\mathbf{X}^{(j)} \in \mathbb{R}^{m \times n_j}$ correspond to the samples in the $j^{th}$ group, $n_j$ is the sample number in the group and $\sum_{j=1}^{b} n_j = n$. Accordingly denote the weight vector as $\mathbf{v} = [\mathbf{v}^{(1)}, \cdots, \mathbf{v}^{(b)}]$, where $\mathbf{v}^{(j)} = (v_1^{(j)}, \cdots, v_{n_j}^{(j)})^T \in [0,1]^{n_j}$. SPLD on one hand needs to assign nonzero weights of $\mathbf{v}$ to easy samples as the conventional SPL, and on the other hand requires to disperse nonzero elements across possibly more groups $\mathbf{v}^{(i)}$ to increase the diversity. Both requirements can be uniformly realized through the following optimization model:

$$\min_{\mathbf{w}, \mathbf{v}} \mathbb{E}(\mathbf{w}, \mathbf{v}; \lambda, \gamma) = \sum_{i=1}^{n} v_i L(y_i, f(\mathbf{x}_i, \mathbf{w})) - \lambda \sum_{i=1}^{n} v_i - \gamma \|\mathbf{v}\|_{2,1}, \text{ s.t. } \mathbf{v} \in [0,1]^n, \quad (3)$$

where $\lambda, \gamma$ are the parameters imposed on the easiness term (the negative $l_1$-norm: $-\|\mathbf{v}\|_1$) and the diversity term (the negative $l_{2,1}$-norm: $-\|\mathbf{v}\|_{2,1}$), respectively. As for the diversity term, we have:

$$-\|\mathbf{v}\|_{2,1} = -\sum_{j=1}^{b} \|\mathbf{v}^{(j)}\|_2. \quad (4)$$

The SPLD introduces a new regularization term in Eq. (3) which consists of two components. One is the negative $l_1$-norm inherited from the conventional SPL, which favors selecting easy over complex examples. The other is the proposed negative $l_{2,1}$-norm, which favors selecting diverse samples residing in more groups. It is well known that the $l_{2,1}$-norm leads to the group-wise sparse representation of $\mathbf{v}$ [14], i.e. non-zero entries of $\mathbf{v}$ tend to be concentrated in a small number of groups. Contrariwise, the negative $l_{2,1}$-norm should have a counter-effect to group-wise sparsity, i.e. nonzero entries of $\mathbf{v}$ tend to be scattered across a large number of groups. In other words, this anti-group-sparsity representation is expected to realize the desired diversity. Note that when each group only contains a single sample, Eq. (3) degenerates to Eq. (1).

Unlike the convex regularization term in Eq. (1) of SPL, the term in the SPLD is non-convex. Consequently, the traditional (sub)gradient-based methods cannot be directly applied to optimizing $\mathbf{v}$. We will discuss an algorithm to resolve this issue in the next subsection.

## 4.2 SPLD Algorithm

Similar as the SPL, the alternative search strategy can be employed for solving Eq. (3). However, a challenge is that optimizing $\mathbf{v}$ with a fixed $\mathbf{w}$ becomes a non-convex problem. We propose a simple yet effective algorithm for extracting the global optimum of this problem, as listed in Algorithm 1. It takes as input the groups of samples, the up-to-date model parameter $\mathbf{w}$, and two self-paced parameters, and outputs the optimal $\mathbf{v}$ of $\min_{\mathbf{v}} \mathbb{E}(\mathbf{w}, \mathbf{v}; \lambda, \gamma)$. The global minimum is proved in the following theorem (see the proof in supplementary materials):

**Theorem 1** *Algorithm 1 attains the global optimum to* $\min_{\mathbf{v}} \mathbb{E}(\mathbf{w}, \mathbf{v})$ *for any given* $\mathbf{w}$ *in linearithmic time.*

As shown, Algorithm 1 selects samples in terms of both the easiness and the diversity. Specifically:

- Samples with $L(y_i, f(\mathbf{x}_i, \mathbf{w})) < \lambda$ will be selected in training ($v_i = 1$) in Step 5. These samples represent the "easy" examples with small losses.

- Samples with $L(y_i, f(\mathbf{x}_i, \mathbf{w})) > \lambda + \gamma$ will not be selected in training ($v_i = 0$) in Step 6. These samples represent the "complex" examples with larger losses.

- Other samples will be selected by comparing their losses to a threshold $\lambda + \frac{\gamma}{\sqrt{i} + \sqrt{i-1}}$, where $i$ is the sample's rank w.r.t. its loss value within its group. The sample with a smaller loss than the threshold will be selected in training. Since the threshold decreases considerably as the rank $i$ grows, Step 5 penalizes samples monotonously selected from the same group.

We study a tractable example that allows for clearer diagnosis in Fig. 2, where each keyframe represents a video sample on the event "Rock Climbing" of the TRECVID MED data [15], and the number below indicates its loss. The samples are clustered into four groups based on the visual similarity. A colored block on the right shows a curriculum selected by Algorithm 1. When $\gamma = 0$,

**Algorithm 1:** Algorithm for Solving $\min_{\mathbf{v}} \mathbb{E}(\mathbf{w}, \mathbf{v}; \lambda, \gamma)$.

**input** : Input dataset $\mathcal{D}$, groups $\mathbf{X}^{(1)}, \cdots, \mathbf{X}^{(b)}$, $\mathbf{w}$, $\lambda$, $\gamma$
**output**: The global solution $\mathbf{v} = (\mathbf{v}^{(1)}, \cdots, \mathbf{v}^{(b)})$ of $\min_{\mathbf{v}} \mathbb{E}(\mathbf{w}, \mathbf{v}; \lambda, \gamma)$.

**1** **for** $j = 1$ **to** $b$ **do** // for each group
**2**     Sort the samples in $\mathbf{X}^{(j)}$ as $(\mathbf{x}_1^{(j)}, \cdots, \mathbf{x}_{n_j}^{(j)})$ in ascending order of their loss values $L$;
**3**     Accordingly, denote the labels and weights of $\mathbf{X}^{(j)}$ as $(y_1^{(j)}, \cdots, y_{n_j}^{(j)})$ and $(v_1^{(j)}, \cdots, v_{n_j}^{(j)})$;
**4**     **for** $i = 1$ **to** $n_j$ **do** // easy samples first
**5**        **if** $L(y_i^{(j)}, f(\mathbf{x}_i^{(j)}, \mathbf{w})) < \lambda + \gamma \frac{1}{\sqrt{i} + \sqrt{i-1}}$ **then** $v_i^{(j)} = 1$ ; // select this sample
**6**        **else** $v_i^{(j)} = 0$; // not select this sample
**7**     **end**
**8** **end**
**9** **return** $\mathbf{v}$

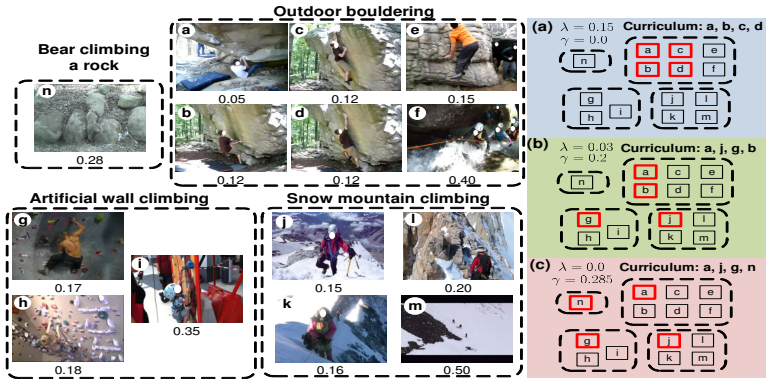

Figure 2: An example on samples selected by Algorithm 1. A colored block denotes a curriculum with given $\lambda$ and $\gamma$, and the bold (red) box indicates the easy sample selected by Algorithm 1.

as shown in Fig. 2(a), SPLD, which is identical to SPL, selects only easy samples (with the smallest losses) from a single cluster. Its curriculum thus includes duplicate samples like $b, c, d$ with the same loss value. When $\lambda \neq 0$ and $\gamma \neq 0$ in Fig. 2(b), SPLD balances the easiness and the diversity, and produces a reasonable and diverse curriculum: $a, j, g, b$. Note that even if there exist 3 duplicate samples $b, c, d$, SPLD only selects one of them due to the decreasing threshold in Step 5 of Algorithm 1. Likewise, samples $e$ and $j$ share the same loss, but only $j$ is selected as it is better in increasing the diversity. In an extreme case where $\lambda = 0$ and $\gamma \neq 0$, as illustrated in Fig. 2(c), SPLD selects only diverse samples, and thus may choose outliers, such as the sample $n$ which is a confusable video about a bear climbing a rock. Therefore, considering both easiness and diversity seems to be more reasonable than considering either one alone. Physically the parameters $\lambda$ and $\gamma$ together correspond to the "age" of the model, where $\lambda$ focuses on easiness whereas $\gamma$ stresses diversity.

As Algorithm 1 finds the optimal $\mathbf{v}$, the alternative search strategy can be readily applied to solving Eq. (3). The details are listed in Algorithm 2. As aforementioned, Step 4 can be implemented using the existing off-the-shelf learning method. Following [6], we initialize $\mathbf{v}$ by setting $v_i = 1$ to randomly selected samples. Following SPL [6], the self-paced parameters are updated by absolute values of $\mu_1, \mu_2$ ($\mu_1, \mu_2 \geq 1$) in Step 6 at the end of every iteration. In practice, it seems more robust by first sorting samples in ascending order of their losses, and then setting the $\lambda, \gamma$ according to the statistics collected from the ranked samples (see the discussion in supplementary materials). According to [6], the alternative search in Algorithm 1 converges as the objective function is monotonically decreasing and is bounded from below.

## 5 Experiments

We present experimental results for the proposed SPLD on two tasks: event detection and action recognition. We demonstrate that our approach significantly outperforms SPL on three real-world challenging datasets. The code is at (http://www.cs.cmu.edu/~lujiang/spld).

---

**Algorithm 2:** Algorithm of Self-Paced Learning with Diversity.

    **input** : Input dataset $\mathcal{D}$, self-pace parameters $\mu_1, \mu_2$
    **output**: Model parameter $\mathbf{w}$

**1**    **if** *no prior clusters exist* **then** cluster the training samples $\mathbf{X}$ into $b$ groups $\mathbf{X}^{(1)}, \cdots, \mathbf{X}^{(b)}$;
**2**    Initialize $\mathbf{v}^*, \lambda, \gamma$ ; // `assign the starting value`
**3**    **while** *not converged* **do**
**4**         Update $\mathbf{w}^* = \arg\min_{\mathbf{w}} \mathbb{E}(\mathbf{w}, \mathbf{v}^*; \lambda, \gamma)$ ; // `train a classification model`
**5**         Update $\mathbf{v}^* = \arg\min_{\mathbf{v}} \mathbb{E}(\mathbf{w}^*, \mathbf{v}; \lambda, \gamma)$ using Algorithm 1; // `select easy & diverse samples`
**6**         $\lambda \leftarrow \mu_1 \lambda$ ; $\gamma \leftarrow \mu_2 \gamma$ ; // `update the learning pace`
**7**    **end**
**8**    **return** $\mathbf{w} = \mathbf{w}^*$

---

SPLD is compared against four baseline methods: 1) **RandomForest** is a robust bootstrap method that trains multiple decision trees using randomly selected samples and features [16]. 2) **AdaBoost** is a classical ensemble approach that combines the sequentially trained "base" classifiers in a weighted fashion [18]. Samples that are misclassified by one base classifier are given greater weight when used to train the next classifier in sequence. 3) **BatchTrain** represents a standard training approach in which a model is trained simultaneously using all samples; 4) **SPL** is a state-of-the-art method that trains models gradually from easy to more complex samples [6]. The baseline methods are a mixture of the well-known and the state-of-the-art methods on training models using sampled data.

### 5.1 Multimedia Event Detection (MED)

**Problem Formulation** Given a collection of videos, the goal of MED is to detect events of interest, e.g. "Birthday Party" and "Parade", solely based on the video content. The task is very challenging due to complex scenes, camera motion, occlusions, etc. [17, 19, 8].

**Dataset** The experiments are conducted on the largest collection on event detection: TRECVID MED13Test, which consists of about 32,000 Internet videos. There are a total of 3,490 videos from 20 complex events, and the rest are background videos. For each event 10 positive examples are given to train a detector, which is tested on about 25,000 videos. The official test split released by NIST (National Institute of Standards and Technology) is used [15].

**Experimental setting** A Deep Convolutional Neural Network is trained on 1.2 million ImageNet challenge images from 1,000 classes [20] to represent each video as a 1,000-dimensional vector. Algorithm 2 is used. By default, the group membership is generated by the spectral clustering, and the number of groups is set to 64. Following [9, 8], LibLinear is used as the solver in Step 4 of Algorithm 2 due to its robust performance on this task. The performance is evaluated using MAP as recommended by NIST. The parameters of all methods are tuned on the same validation set.

Table 1 lists the overall MAP comparison. To reduce the influence brought by initialization, we repeated experiments of SPL and SPLD 10 times with random starting values, and report the best run and the mean (with the 95% confidence interval) of the 10 runs. The proposed SPLD outperforms all baseline methods with statistically significant differences at the $p$-value level of 0.05, according to the paired t-test. It is worth emphasizing that MED is very challenging [15] and 26% relative (2.5 absolute) improvement over SPL is a notable gain. SPLD outperforms other baselines on both the best run and the 10 runs average. RandomForest and AdaBoost yield poorer performance. This observation agrees with the study in literature [15, 9] that SVM is more robust on event detection.

Table 1: MAP (x100) comparison with the baseline methods on MED.

| Run Name | RandomForest | AdaBoost | BatchTrain | SPL | SPLD |
|---|---|---|---|---|---|
| Best Run | 3.0 | 2.8 | 8.3 | 9.6 | **12.1** |
| 10 Runs Average | 3.0 | 2.8 | 8.3 | 8.6±0.42 | **9.8±0.45** |

BatchTrain, SPL and SPLD are all performed using SVM. Regarding the best run, SPL boosts the MAP of the BatchTrain by a relative 15.6% (absolute 1.3%). SPLD yields another 26% (absolute 2.5%) over SPL. The MAP gain suggests that optimizing objectives with the diversity is inclined to attain a better solution. Fig. 3 plots the validation and test AP on three representative events. As illustrated, SPLD attains a better solution within fewer iterations than SPL, e.g. in Fig. 3(a) SPLD obtains the best test AP (0.14) by 6 iterations as opposed to AP (0.12) by 11 iterations in

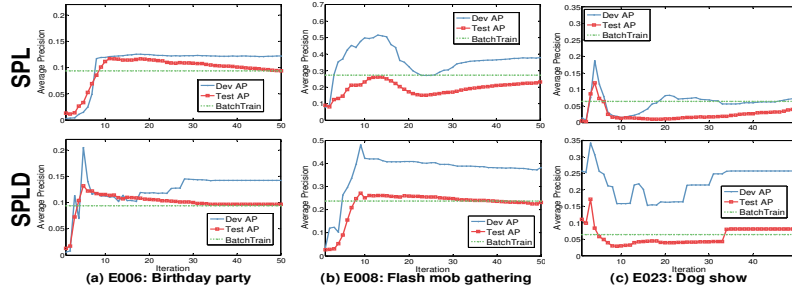

Figure 3: The validation and test AP in different iterations. Top row plots the SPL result and bottom shows the proposed SPLD result. The $x$-axis represents the iteration in training. The blue solid curve (Dev AP) denotes the AP on the validation set, the red one marked by squares (Test AP) denotes the AP on the test set, and the green dashed curve denotes the Test AP of BatchTrain which remains the same across iterations.

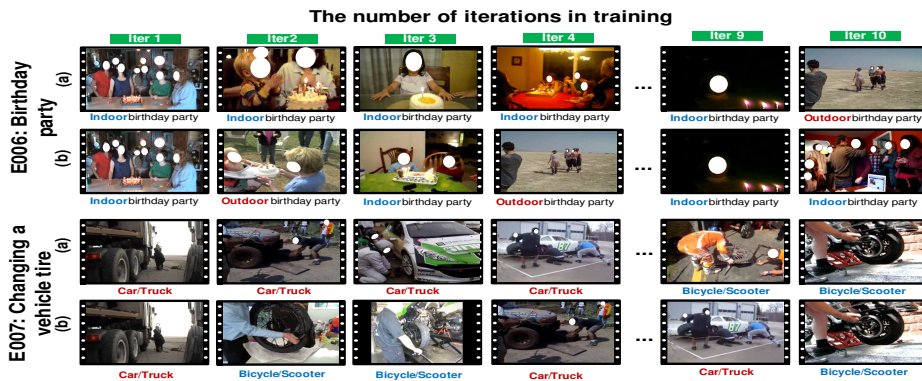

Figure 4: Comparison of positive samples used in each iteration by (a) SPL (b) SPLD.

SPL. Studies [1, 6] have shown that SPL converges fast, while this observation further suggests that SPLD may lead to an even faster convergence. We hypothesize that it is because the diverse samples learned in the early iterations in SPLD tend to be more informative. The best Test APs of both SPL and SPLD are better than BatchTrain, which is consistent with the observation in [5] that removing some samples may be beneficial in training a better detector. As shown, Dev AP and Test AP share a similar pattern justifying the rationale for parameters tuning on the validation set.

Fig. 4 plots the curriculum generated by SPL and SPLD in a first few iterations on two representative events. As we see, SPL tends to select easy samples similar to what it has already learned, whereas SPLD selects samples that are both easy and diverse to the model. For example, for the event "E006 Birthday Party", SPL keeps selecting indoor scenes due to the sample learned in the first place. However, the samples learned by SPLD are a mixture of indoor and outdoor birthday parties. For the complex samples, both methods leave them to the last iterations, e.g. the 10th video in "E007".

## 5.2 Action Recognition

**Problem Formulation** The goal is to recognize human actions in videos.

**Datasets** Two representative datasets are used: Hollywood2 was collected from 69 different Hollywood movies [21]. It contains 1,707 videos belonging to 12 actions, splitting into a training set (823 videos) and a test set (884 videos). Olympic Sports consists of athletes practicing different sports collected from YouTube [22]. There are 16 sports actions from 783 clips. We use 649 for training and 134 for testing as recommended in [22].

**Experimental setting** The improved dense trajectory feature is extracted and further represented by the fisher vector [23, 24]. A similar setting discussed in Section 5.1 is applied, except that the groups are generated by K-means ($K$=128).

Table 2 lists the MAP comparison on the two datasets. A similar pattern can be observed that SPLD outperforms SPL and other baseline methods with statistically significant differences. We then compare our MAP with the state-of-the-art MAP in Table 3. Indeed, this comparison may be

Table 2: MAP (x100) comparison with the baseline methods on Hollywood2 and Olympic Sports.

| Run Name | RandomForest | AdaBoost | BatchTrain | SPL | SPLD |
|---|---|---|---|---|---|
| Hollywood2 | 28.20 | 41.14 | 58.16 | 63.72 | **66.65** |
| Olympic Sports | 63.32 | 69.25 | 90.61 | 90.83 | **93.11** |

less fair since the features are different in different methods. Nevertheless, with the help of SPLD, we are able to achieve the best MAP reported so far on both datasets. Note that the MAPs in Table 3 are obtained by recent and very competitive methods on action recognition. This improvement confirms the assumption that considering diversity in learning is instrumental.

Table 3: Comparison of SPLD to the state-of-the-art on Hollywood2 and Olympic Sports

| Hollywood2 | | Olympic Sports | |
|---|---|---|---|
| Vig et al. 2012 [25] | 59.4% | Brendel et al. 2011 [28] | 73.7% |
| Jiang et al. 2012 [26] | 59.5% | Jiang et al. 2012 [26] | 80.6% |
| Jain et al. 2013 [27] | 62.5% | Gaidon et al. 2012 [29] | 82.7% |
| Wang et al. 2013 [23] | 64.3% | Wang et al. 2013 [23] | 91.2% |
| **SPLD** | **66.7%** | **SPLD** | **93.1%** |

## 5.3 Sensitivity Study

We conduct experiments using different number of groups generated by two clustering algorithm: K-means and Spectral Clustering. Each experiment is fully tuned under the given #groups and the clustering algorithm, and the best run is reported in Table 4. The results suggest that SPLD is relatively insensitive to the clustering method and the given group numbers. We hypothesize that SPLD may not improve SPL in the cases where the assumption in Section 4.1 is violated, and the given groups, e.g. random clusters, cannot reflect the latent variousness in data.

Table 4: MAP (x100) comparison of different clustering algorithms and #clusters.

| Dataset | SPL | Clustering | #Groups=32 | #Groups=64 | #Groups=128 | #Groups=256 |
|---|---|---|---|---|---|---|
| MED | 8.6±0.42 | K-means | 9.16±0.31 | 9.20±0.36 | 9.25±0.32 | 9.03±0.28 |
| | | Spectral | 9.29±0.42 | **9.79±0.45** | 9.22±0.41 | 9.38±0.43 |
| Hollywood2 | 63.72 | K-means | 66.372 | 66.358 | 66.653 | 66.365 |
| | | Spectral | 66.639 | 66.504 | 66.264 | **66.709** |
| Olympic | 90.83 | K-means | 91.86 | 92.37 | 93.11 | 92.65 |
| | | Spectral | 91.08 | 92.51 | **93.25** | 92.54 |

## 6 Conclusion

We advanced the frontier of the self-paced learning by proposing a novel idea that considers both easiness and diversity in learning. We introduced a non-convex regularization term that favors selecting both easy and diverse samples. The proposed regularization term is general and can be applied to various problems. We proposed a linearithmic algorithm that finds the global optimum of this non-convex problem on updating the samples to be included. Using three real-world datasets, we showed that the proposed SPLD outperforms the state-of-the-art approaches.

Possible directions for future work may include studying the diversity for samples in the mixture model, e.g. mixtures of Gaussians, in which a sample is assigned to a mixture of clusters. Another possible direction would be studying assigning reliable starting values for SPL/SPLD.

**Acknowledgments**

This work was partially supported by Intelligence Advanced Research Projects Activity (IARPA) via Department of Interior National Business Center contract number D11PC20068. Deyu Meng was partially supported by 973 Program of China (3202013CB329404) and the NSFC project (61373114). The U.S. Government is authorized to reproduce and distribute reprints for Governmental purposes notwithstanding any copyright annotation thereon. Disclaimer: The views and conclusions contained herein are those of the authors and should not be interpreted as necessarily representing the official policies or endorsements, either expressed or implied, of IARPA, DoI/NBC, or the U.S. Government.

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
