[Supplementary Material]

# Supplementary Materials: Self-Paced Learning with Diversity

**Lu Jiang**[1], **Deyu Meng**[1,2], **Shoou-I Yu**[1], **Zhenzhong Lan**[1], **Shiguang Shan**[1,3],
**Alexander G. Hauptmann**[1]

[1]School of Computer Science, Carnegie Mellon University
[2]School of Mathematics and Statistics, Xi'an Jiaotong University
[3]Institute of Computing Technology, Chinese Academy of Sciences
lujiang@cs.cmu.edu, dymeng@mail.xjtu.edu.cn
{iyu, lanzhzh}@cs.cmu.edu, sgshan@ict.ac.cn, alex@cs.cmu.edu

## Abstract

This is the supplementary material for the paper entitled "Self-Paced Learning with Diversity". The material is organized as follows: Section 1 gives the proof of Theorem 1. Section 2.1 and Section 2.2 present the detailed experimental settings and results on the MED (Multimedia Event Detection) dataset. Section 2.3 and Section 2.4 present the settings and detailed results on the Hollywood2 and Olympic datasets. Section 3 briefly discusses our practical lessons and the observed deficiency of the SPL/SPLD models.

## 1 Proof of Theorem 1

We present the proof of Theorem 1 in the paper. Given the training dataset $\mathcal{D} = \{(\mathbf{x}_1, y_1), \cdots, (\mathbf{x}_n, y_n)\}$, where $\mathbf{x}_i \in \mathbb{R}^m$ denotes the $i^{th}$ observed sample and $y_i$ denotes its label. Assume that the training samples $\mathbf{X} = [\mathbf{x}_1, \cdots, \mathbf{x}_n]$ are with $b$ groups: $\mathbf{X}^{(1)}, \cdots, \mathbf{X}^{(b)}$, where $\mathbf{X}^{(j)} = (\mathbf{x}_1^{(j)}, \cdots, \mathbf{x}_{n_j}^{(j)}) \in \mathbb{R}^{m \times n_j}$ corresponds to samples in the $j^{th}$ group, $n_j$ is the sample number in this group and $\sum_{j=1}^{b} n_j = n$. Accordingly, denote the weight vector as $\mathbf{v} = [\mathbf{v}^{(1)}, \cdots, \mathbf{v}^{(b)}]$, where $\mathbf{v}^{(j)} = (v_1^{(j)}, \cdots, v_{n_j}^{(j)})^T \in \mathbb{R}^{n_j}$. The following theorem proves that Algorithm 1 can get the global solution of the following non-convex optimization problem:

$$\min_{\mathbf{v} \in [0,1]^n} \mathbb{E}(\mathbf{w}, \mathbf{v}; \lambda, \gamma) = \sum_{i=1}^{n} v_i L(y_i, f(\mathbf{x}_i, \mathbf{w})) - \lambda \sum_{i=1}^{n} v_i - \gamma \|\mathbf{v}\|_{2,1}, \qquad (1)$$

where $L(y_i, f(\mathbf{x}_i, \mathbf{w}))$ denotes the loss function which calculates the cost between the ground truth label $y_i$ and the estimated label $f(\mathbf{x}_i, \mathbf{w})$, and the $l_{2,1}$-norm $\|\mathbf{v}\|_{2,1}$ is the group sparsity of $\mathbf{v}$:

$$\|\mathbf{v}\|_{2,1} = \sum_{j=1}^{b} \|\mathbf{v}^{(j)}\|_2.$$

For convenience we briefly rewrite $\mathbb{E}(\mathbf{w}, \mathbf{v}; \lambda, \gamma)$ and $L(y_i, f(\mathbf{x}_i, \mathbf{w}))$ as $\mathbb{E}(\mathbf{v})$ and $L_i$, respectively, throughout this material.

**Theorem 1** *The weight vector $\mathbf{v}^*$ outputted from Algorithm 1 attains the global optimal solution of the optimization problem (1), i.e.,*

$$\mathbf{v}^* = \arg \min_{\mathbf{v} \in [0,1]^n} \mathbb{E}(\mathbf{v}).$$

**Proof 1** *The objective function of (1) can be reformulated as the following decoupling forms based on the data cluster information:*

$$\mathbb{E}(\mathbf{v}) = \sum_{j=1}^{b} E(\mathbf{v}^{(j)}), \tag{2}$$

*where*

$$E(\mathbf{v}^{(j)}) = \sum_{i=1}^{n_j} v_i^{(j)} L_i^{(j)} - \lambda \sum_{i=1}^{n_j} v_i^{(j)} - \gamma \|\mathbf{v}^{(j)}\|_2, \tag{3}$$

*where $L_i^{(j)}$ represents the loss value of $\mathbf{x}_i^{(j)}$. It is easy to see that the original problem (1) can be equivalently decomposed as a series of the following sub-optimization problems ($j = 1, \cdots, b$):*

$$\mathbf{v}^{(j)*} = \arg \min_{\mathbf{v}^{(j)} \in [0,1]^{n_j}} \mathbb{E}(\mathbf{v}^{(j)}). \tag{4}$$

*$E(\mathbf{v}^{(j)})$ defined in Eq. (3) is a concave function since its first and second terms are linear, and the third term is the negative $l_{2,1}$ norm, whose positive form is a commonly utilized convex regularizer. It is well known that the minimum solution of a concave function over a polytope can be obtained at its vertices [1]. In other words, for the optimization problem (4), it holds that its optimal solution $\mathbf{v}^{(j)*} \in \{0,1\}^{n_j}$, i.e.,*

$$\mathbf{v}^{(j)*} = \arg \min_{\mathbf{v}^{(j)} \in \{0,1\}^{n_j}} \mathbb{E}(\mathbf{v}^{(j)}). \tag{5}$$

*For $k = 1, \cdots, n_j$, let's denote*

$$\mathbf{v}^{(j)}(k) = \arg \min_{\substack{\mathbf{v}^{(j)} \in \{0,1\}^{n_j} \\ \|\mathbf{v}^{(j)}\|_0 = k}} \mathbb{E}(\mathbf{v}^{(j)}). \tag{6}$$

*This means that $\mathbf{v}^{(j)}(k)$ is the optimum of (4) if it is further constrained to be with $k$ nonzero entries. It is then easy to deduce that*

$$\mathbf{v}^{(j)*} = \arg \min_{\mathbf{v}^{(j)}(k)} \mathbb{E}(\mathbf{v}^{(j)}(k)). \tag{7}$$

*That is, the optimal solution $\mathbf{v}^{(j)*}$ of (4) can be achieved among $\mathbf{v}^{(j)}(1), \cdots, \mathbf{v}^{(j)}(n_j)$ at which the minimal objective value is attained.*

*Without loss of generality, we assume that the samples $(\mathbf{x}_1^{(j)}, \cdots, \mathbf{x}_{n_j}^{(j)})$ in the $j^{th}$ cluster are arranged in the ascending order of their loss values $L_i^{(j)}$. Then for the optimization problem (6), we can get that*

$$\min_{\substack{\mathbf{v}^{(j)} \in \{0,1\}^{n_j} \\ \|\mathbf{v}^{(j)}\|_0 = k}} \mathbb{E}(\mathbf{v}^{(j)}) = \sum_{i=1}^{n_j} v_i^{(j)} L_i^{(j)} - \lambda \sum_{i=1}^{n_j} v_i^{(j)} - \gamma \|\mathbf{v}^{(j)}\|_2$$

$$\Leftrightarrow \min_{\substack{\mathbf{v}^{(j)} \in \{0,1\}^{n_j} \\ \|\mathbf{v}^{(j)}\|_0 = k}} \sum_{i=1}^{n_j} v_i^{(j)} L_i^{(j)},$$

*since the last two terms in $\mathbb{E}(\mathbf{v}^{(j)})$ are with constant values under the constraint. Then it is easy to get that the optimal solution $\mathbf{v}^{(j)}(k)$ of (6) is attained by setting its $k$ entries corresponding to the $k$ smallest loss values $L_i^{(j)}$ (i.e., the first $k$ entries of $\mathbf{v}^{(j)}(k)$) as 1 while others as 0, and the minimal objective value is*

$$E(\mathbf{v}^{(j)}(k)) = \sum_{i=1}^{k} v_i^{(j)} L_i^{(j)} - \lambda k - \gamma \sqrt{k}. \tag{8}$$

*Then let's calculate the difference between any two adjacent elements in the sequence* $E(\mathbf{v}^{(j)}(1)), \cdots , E(\mathbf{v}^{(j)}(n_j))$*:*

$$\begin{aligned} diff_k &= E(\mathbf{v}^{(j)}(k+1)) - E(\mathbf{v}^{(j)}(k)) \\ &= L_{k+1}^{(j)} - \lambda - \gamma(\sqrt{k+1} - \sqrt{k}) \\ &= L_{k+1}^{(j)} - (\lambda + \gamma \frac{1}{\sqrt{k+1}+\sqrt{k}}). \end{aligned}$$

*Since $L_k^{(j)}$ (with respect to $k$) is a monotonically increasing sequence while $\lambda + \gamma\frac{1}{\sqrt{k+1}+\sqrt{k}}$ is a monotonically decreasing sequence, $diff_k$ is a monotonically increasing sequence. Denote $k^*$ as the index where its first positive value is attained (if $diff_k \le 0$ for all $k = 1, \cdots , n_j - 1$, $k^* = n_j$). Then it is easy to get that $E(\mathbf{v}^{(j)}(k))$ is monotonically decreasing until $k = k^*$ and then it starts to be monotonically increasing. This means that $E(\mathbf{v}^{(j)}(k^*))$ gets the minimum among all $E(\mathbf{v}^{(j)}(1)), \cdots , E(\mathbf{v}^{(j)}(n_j))$. Based on (7), we know that the global optimum $\mathbf{v}^{(j)*}$ of (4) is attained at $\mathbf{v}^{(j)}(k^*)$.*

*By independently calculating the optimum $\mathbf{v}^{(j)*}$ for each cluster and then combining them, the global optimal solution $\mathbf{v}^*$ of (1) can then be calculated. This corresponds to the process of our proposed Algorithm 1.*

*The most computational complex step in the above derivation is the sort of $n_j$ $(1 \le j \le b)$ samples. Since $n_j < n$, the average-case complexity is thus upper bounded by $O(n \log n)$, assuming that the quick sort algorithm is used.*

*The proof is completed.* ∎

## 2    Experiments

### 2.1    Experimental Setting on Event Detection

The TRECVID MED13Test dataset [1] is by far the largest collection on event detection. The set consists of about 32,000 Internet videos. There are a total of 3,490 videos from 20 complex events, e.g. "Birthday party" and "Parade". The rest are background videos. For each event 10 positive examples are given to train a detector, which is tested on about 25,000 videos. The official test split released by NIST (National Institute of Standards and Technology) is used [2].

The R language 3.0.1 [3] [2] is used in our implementation. The code and the data are at this page [3]. The proposed SPLD is compared against the four baseline methods: RandomForest [4], AdaBoost [5], BatchTrain and SPL (Self-paced Learning) [6].

A Deep Convolutional Neural Network [7] [4] is trained on 1.2 million ImageNet challenge images from 1,000 classes [8] to represent each video as a 1,000-dimensional vector [9]. For BatchTrain, SPL and SPLD, studies have shown that $\chi^2$ kernel [10, 2, 9] usually performs better than the linear kernel. To incorporate the $\chi^2$ kernel into these models, we map the 1,000 dimensional space into a 5,000 dimension space by the explicit feature mapping using the additive $\chi^2$ kernel [11] [5]. To run the baseline methods and the proposed method efficiently, the feature dimension is further reduced to 512 by principal component analysis (PCA), because the 512-dimensional features yield very similar results as the 5,000-dimensional features. The same set of features is used across all methods.

The "randomForest" package is used to train the random forest classifier [6]. Two parameters are tuned on the validation set: the number of trees ($\{32, 64, 128, 256, 1024, 2048\}$) and the number variables sampled as candidates at each split ($\{16, 32, 64, 128\}$). In total, 24 sets of experiments are conducted, and the run with the best MAP (0.03) is selected in the baseline comparison.

The "ada" package is used to train the AdaBoost classifier [12] [7]. Two parameters are tuned on the validation set: the loss in the objective function {exponential, logistic}, and the number of iterations in $\{10, 30, 50, 70, 90\}$. In total, 10 sets of experiments are conducted, and the run with the best MAP (0.028) is selected in the baseline comparison.

The "Liblinear" package [13] is used to train BatchTrain [8]. Two parameters are tuned: the tolerance parameter $C = \{0.1, 1, 10\}$, the 6 loss functions ({$L_2$-regularized $L_1$-loss SVM, $L_2$-regularized $L_2$-loss SVM, $L_2$-regularized logistic regression, $L_1$-regularized logistic regression, etc.}). The bias term is set to 1.0. The run with the best MAP (0.083) is selected in the baseline comparison ($L_2$-regularized logistic regression, bias term is set to 1.0 and $C = 1.0$).

The SPL algorithm in [6] is implemented based on the SVM classier by the "Liblinear" package using the same $C$, bias term and the loss function as BatchTrain. The parameter $\lambda$ is tuned in terms of the rank rather than the absolute value. That is instead of specifying the absolute value of $\lambda$ in each iteration, we specify the number of samples to be included in each iteration, and then calculate $\lambda$ accordingly. For example, suppose 5 samples are needed to be selected for the current iteration; we first sort samples in ascending order of their losses, and set $\lambda$ to be the loss of the 6th sample so that only the top 5 samples will be selected. This implementation is theoretically consistent with the algorithm in [6]. Empirically we found setting $\lambda$ by the statistics collected from ranked samples is more robust than by absolute values for both SPL and SPLD. This is because setting by absolute values can result in selecting too many or too few samples. Since the training set is extremely unbalanced which contains only 10 positive samples but around 5000 negative samples. Two parameters $\lambda_+, \lambda_-$ are incorporated for positive and negative samples, respectively.

Algorithm 1 and Algorithm 2 in the paper are used in implementing SPLD. The group membership in Algorithm 1 is generated by two clustering algorithms on the reduced-dimension feature. The "stat" package is used to implement $K$-means. "kernlab" package is used in the spectral clustering [14] implementation [9]. The number of groups in both algorithms are set to {64,128,256}. By default, the best configuration on the validation set (64 clusters generated by the spectral clustering) is used in the baseline comparison. Likewise, the parameters $\lambda, \gamma$ are set according to the ranked samples, and then the absolute values of $\lambda, \gamma$ are calculated accordingly. These two parameters are fully tuned in the same manner as in SPL, where the line search strategy is used. In all experiments, the multi-class classification is conducted using the "one-versus-all" scheme. A set of randomly selected videos including 10 positive videos (neither from the training nor the test set) are used as the validation set. For a fair comparison, the parameters of all methods are tuned on the same validation set. For example, in SPL and SPLD, the best iteration to train the final model is searched on the validation set. The final model of both SPL and SPLD only incorporates a subset of the given training set.

In SPL and SPLD, following [6], we initialize $\mathbf{v}$ by setting $v_i = 1$ to a number of randomly selected samples. We observed both SPL and SPCL may be unstable to the random starting values. We repeat the experiments 10 times with different starting values. The 10-run average and the best run are reported.

## 2.2 Detailed Results on MED

Table 1 lists the AP for each event in the MED dataset. As we see, the improvement is consistent across events. SPLD obtains the best AP on 13 out of 20 events, and obtains the top 2 AP on 17 out of 20 events. According to the paired t-test (one tail), the improvement of SPLD over all baseline is statistically significant at the $p$-value level of 0.02. It is worth mentioning that the task is very challenging [15, 16, 17], and the MAP of our BatchTrain result is comparable to the state-of-the-art result on the MED using 10 samples and a single type of feature [2].

Fig. 1 plots the validation and the test AP on three representative events, in which the $x$-axis represents the iteration in training. The blue solid curve (Dev MAP) denotes the MAP on the validation set, the red one marked by squares (Test AP) denotes the MAP on the test set, and the green dashed curve denotes the AP of BatchTrain which remains the same across iterations. As illustrated in

Table 1: Performance comparison with the baseline methods on TRECVID MED. The best AP for each event is marked in bold.

| Event ID & Name | RandomForest | AdaBoost | BatchTrain | SPL | SPLD |
|---|---|---|---|---|---|
| E006: Birthday party | 1.498 | 1.696 | 9.411 | 11.659 | **13.202** |
| E007: Changing a vehicle tire | 7.819 | 4.541 | 19.148 | **20.232** | 19.723 |
| E008: Flash mob gathering | 3.357 | 15.147 | 23.673 | 26.108 | **27.149** |
| E009: Getting a vehicle unstuck | 7.711 | 10.83 | 11.349 | 11.516 | **28.845** |
| E010: Grooming an animal | 4.065 | 1.522 | 5.830 | 7.437 | **8.996** |
| E011: Making a sandwich | 2.497 | 1.076 | 9.347 | 8.364 | **12.594** |
| E012: Parade | 3.767 | 3.951 | 16.255 | **19.553** | **19.553** |
| E013: Parkour | 7.299 | 9.012 | 27.403 | 27.403 | **30.105** |
| E014: Repairing an appliance | 3.695 | 0.581 | 18.237 | 18.237 | **25.898** |
| E015: Working on a sewing project | 0.497 | 0.342 | 2.482 | 2.740 | **2.943** |
| E021: Attempting a bike trick | 1.076 | 0.402 | **3.965** | **3.965** | 3.806 |
| E022: Cleaning an appliance | 0.476 | 0.499 | 0.932 | **5.964** | 1.156 |
| E023: Dog show | 8.696 | 1.497 | 6.371 | 12.018 | **17.103** |
| E024: Giving directions to a location | 0.205 | 0.218 | 0.548 | **0.847** | 0.701 |
| E025: Marriage proposal | 0.180 | 0.127 | 0.313 | **0.590** | 0.522 |
| E026: Renovating a home | 0.188 | 0.246 | 1.368 | 1.552 | **3.957** |
| E027: Rock climbing | 4.381 | 3.284 | 3.288 | 3.288 | **7.718** |
| E028: Town hall meeting | 0.572 | 0.295 | 2.406 | **5.964** | 2.848 |
| E029: Winning a race without a vehicle | 0.650 | 0.730 | 1.953 | 2.033 | **14.783** |
| E030: Working on a metal crafts project | **2.360** | 0.410 | 0.861 | 1.565 | 0.641 |
| MAP (x100) | 3.049 | 2.820 | 8.257 | 9.552 | **12.112** |

Figure 1: The validation and test AP in different iterations. Top row plots the SPL result and bottom shows the proposed SPLD result. The $x$-axis represents the iteration in training. The blue solid curve (Dev AP) denotes the AP on the validation set, the red one marked by squares (Test AP) denotes the AP on the test set, and the green dashed curve denotes the Test AP of BatchTrain which remains the same across iterations.

Fig. 1, SPLD attains a better solution within fewer iterations than SPL, e.g. in Fig. 1(a) SPLD obtains the best test AP (0.14) by 6 iterations as opposed to AP (0.12) by 11 iterations in SPL. We hypothesize that it is because the diverse samples learned in the early iterations in SPLD tend to be more informative. This result substantiates the claim that considering diversity in SPL is beneficial. The best Test APs of both SPL and SPLD are better than BatchTrain. This observation is consistent with the one in [18] that removing some samples may be beneficial in training a better detector. The Dev APs are used to tune the best iteration to train models. As shown, Dev AP and Test AP share a similar pattern justifying the rationale for parameters tuning on the validation set.

## 2.3 Experimental Setting on Action Recognition

Hollywood2 [10] and Olympic Sports [11] are two representative datasets on action recognition. Hollywood2 were collected from 69 different Hollywood movies [19]. It contains 1,707 videos belonging to 12 actions, split into a training set (823 videos) and a test set (884 videos). Olympic Sports consists of athletes practicing different sports collected from YouTube [20]. There are 16 sports actions (such as "High-jump" and "Basketball lay-up") from 783 clips. We use 649 for training and 134 for testing as recommended in [20].

Following [21], the improved dense trajectory features are extracted and represented by fisher vector encoding [22]. The spatial and temporal extension discussed in [23] is applied on the improved dense trajectory. In Hollywood2, the spatial tiling [24] is also applied. As a result, the dimension of the final dense trajectory feature for Hollywood2 is 116,736, and for Olympic Sports is 350,208. To accelerate the experiments, we precalculate the linear kernel matrix, and the kernel SVM implementation in "kernlab" is used in BatchTrain, SPL and SPLD [12]. In Adaboost and RandomForest, since the precomputed kernel is not applicable, it is extremely time consuming to run experiments in the original high-dimensional feature space. Therefore we use PCA to reduce the largest possible dimension (the total number of samples) for Adaboost and RandomForest.

The experiments on these datasets are configured in the same setting discussed in Section 2.1, except the following points. A randomly selected training set is used as the validation set on Olympic Sports, and all training samples are used on Hollywood2. The parameters of all methods on the dataset are tuned on the same validation set. As recommended in [21], for a fair comparison, the $C$ in SVM is fixed across BatchTrain, SPL and SPLD. By default, the $K$-means clustering is used, and the number of clusters is set to 128. We have better results with spectral clustering but this configuration is good enough for the baseline comparison. To compare with the MAP in literature such as [21], only the best MAP found in the experiments is reported. Besides, since many actions in Olympic Sports are easy with 100% Average Precision (AP), there exist multiple iterations with the same dev AP. In other words, ties often occur in parameter tuning on Olympic Sports (see Fig. 3 for some examples). We use some heuristic rules to break ties in the Olympic Sports dataset.

## 2.4 Detailed Results on Hollywood2 and Olympic Sports

Table 2 lists the AP for each action on the Hollywood2 dataset. As on the MED dataset, the improvement is consistent across actions. SPLD obtains the best AP on 9 out of 12 actions (on two actions marked in italic SPLD ties with SPL and BatchTrain). According to the paired t-test, the improvement of SPLD over all baseline is statistically significant at the $p$-value level of $0.1$.

Table 2: Performance comparison with the baseline methods on Hollywood2. The best AP is marked in bold. The italic number indicates that the best AP is shared by more than one methods.

| Action ID & Name | RandomForest | AdaBoost | BatchTrain | SPL | SPLD |
|---|---|---|---|---|---|
| H01: AnswerPhone | 20.492 | 18.350 | 18.775 | 33.719 | **39.462** |
| H02: DriveCar | 68.825 | 88.729 | *95.790* | *95.790* | *95.790* |
| H03: Eat | 11.041 | 17.889 | *71.750* | *71.750* | *71.750* |
| H04: FightPerson | 48.067 | 69.928 | 81.960 | 81.960 | **81.961** |
| H05: GetOutCar | 11.559 | 28.974 | 62.787 | 62.786 | **62.787** |
| H06: HandShake | 8.206 | 6.042 | 42.988 | 42.982 | **46.297** |
| H07: HugPerson | 10.846 | 23.362 | 16.716 | 33.595 | **58.006** |
| H08: Kiss | 40.299 | 46.424 | **63.340** | 62.420 | 60.885 |
| H09: Run | 46.916 | 70.865 | **85.751** | 80.524 | 81.825 |
| H10: SitDown | 30.832 | 66.650 | 53.595 | 81.582 | **81.703** |
| H11: SitUp | 5.307 | 7.437 | 38.860 | 38.870 | **38.874** |
| H12: StandUp | 35.969 | 48.997 | 65.657 | **80.647** | 80.499 |
| MAP (x100) | 28.196 | 41.137 | 58.164 | 63.885 | **66.653** |

Figure 2: The validation AP (Dev AP) and the test AP (Test AP) of different iterations on Hollywood2.

Table 3 lists the AP for each action on the Olympic Sports dataset. A similar patten can be observed that the improvement is consistent across actions. SPLD obtains the best AP across all 16 actions, though on 11 actions the best AP is shared with other baseline methods). The reason is because actions in this dataset are relatively easy to recognize. Since the AP is perfect (100.00) or nearly perfect on many events, SPLD may not further improve the number. According to the paired t-test, the improvement of SPLD over all baseline is statistically significant at the $p$-value level of $0.07$.

Fig. 2 and Fig. 3 plot the validation and test AP on Hollywood2 and Olympic Sports, respectively. For most of the actions, SPLD attains the best solution within fewer iterations than SPL, e.g. in Fig. 3(a) SPLD obtains the perfect test AP (1.0) by 14 iterations as opposed to by 22 iterations in SPL. An interesting observation is that in Fig. 2(a) and Fig. 2(b), one can observe a fluctuation of AP in SPL after it reaches the best Test AP. The fluctuation suggests the samples learned are informative and as a result the model is changed. This type of fluctuation can be only observed in the early iterations of SPLD, suggesting that SPLD selects informative samples earlier. This result could be another evidence for that considering diversity in the self-paced learning is beneficial.

Table 3: Performance comparison with the baseline methods on Olympic Sports. The best AP is marked in bold. The italic number indicates that the best AP is shared by more than one methods.

| Event ID & Name | RandomForest | AdaBoost | BatchTrain | SPL | SPLD |
|---|---|---|---|---|---|
| O01: Basketball layup | 75.80 | 94.88 | *100.00* | *100.00* | *100.00* |
| O02: Bowling | 76.30 | 82.78 | *90.53* | *90.53* | *90.53* |
| O03: Clean and jerk | 69.24 | 97.14 | 98.09 | 98.09 | **100.00** |
| O04: Discus throw | 52.47 | 56.48 | *91.48* | *91.48* | *91.48* |
| O05: Diving platform 10m | *100.00* | *100.00* | *100.00* | *100.00* | *100.00* |
| O06: Diving springboard 3m | 83.39 | 98.61 | *100.00* | *100.00* | *100.00* |
| O07: Hammer throw | 78.94 | 81.22 | *98.61* | *98.61* | *98.61* |
| O08: High jump | 60.18 | 44.31 | 78.16 | 78.16 | **80.30** |
| O09: Javelin throw | 77.86 | 32.92 | *100.00* | *100.00* | *100.00* |
| O10: Long jump | 56.23 | 83.83 | 80.00 | 83.57 | **88.33** |
| O11 Pole vault | 57.02 | 53.75 | *100.00* | *100.00* | *100.00* |
| O12: Shot put | 51.69 | 57.29 | *90.43* | *90.43* | *90.43* |
| O13: Snatch | 89.89 | 88.70 | *92.37* | *92.37* | *92.37* |
| O14: Tennis serve | 16.75 | 45.11 | *96.83* | *96.83* | *96.83* |
| O15: Triple jump | 6.73 | 10.14 | 53.06 | 53.06 | **77.17** |
| O16: Vault | 60.65 | 80.77 | 80.14 | 80.14 | **83.66** |
| MAP (x100) | 63.32 | 69.25 | 90.61 | 90.83 | **93.11** |

## 3 Discussions

We employed two engineering tricks in the implementation. First, the parameters $\lambda$ and $\gamma$ were tuned by the statistics collected from the ranked samples, as opposed to by the absolute values. This strategy avoids selecting too many or too few samples at a single iteration. Second, for unbalanced

Figure 3: The validation AP (Dev AP) and the test AP (Test AP) of different iterations on Olympic Sports.

datasets, such as the ones in our experiments, two sets of parameter $\lambda$ were introduced: $\lambda_+$ for positive and $\lambda_-$ for negative samples in order to pace positive and negative separately. The above implementation, which is consistent with our theory in the paper, behaved more robustly in practice.

In SPL/SPLD, for non-convex loss function $L$ in the off-the-shelf model, the sequential self-paced steps affect the final model. However, for the convex loss function, the same training sets end up with the same model, irrespective of iterative self-paced steps. In our experiments, the loss functions were all convex functions as they are generally regarded as the best model for event/action detection [25, 21]. The converged final SPL/SPLD models only contained a subset of training samples but performed better than the model trained on the whole training set. This is consistent with the observation in [18] that not all samples are valuable in training detectors. As other machine learning methods, SPL/SPLD requires a validation set that follows the same underlying distribution of the test set, in order to tune the parameters. Intuitively, this is analogous to the mock exam whose purposes are to let students realize how well they would perform on the real test data, and more importantly have a better idea of what to study.

We observed two limitations for the current SPL/SPLD models. First, the sample weight variable $v_i$ in SPL/SPLD can only have binary values, which may be less reasonable for the applications that need to discriminate importance of samples. The real-valued weights of self-paced learning were studied in [25]. However, the models in [25] do not support learning with diversity. Second, the performance of SPL/SPLD may be unstable to the random starting values. This phenomenon can be intuitively explained in the context of education, it is impossible for students to predetermine what to learn before they actually learning anything. This problem may be alleviated by incorporating prior knowledge from external sources.

### Acknowledgments

This work was partially supported by Intelligence Advanced Research Projects Activity (IARPA) via Department of Interior National Business Center contract number D11PC20068. Deyu Meng was partially supported by 973 Program of China (3202013CB329404) and the NSFC project (61373114). The U.S. Government is authorized to reproduce and distribute reprints for Governmental purposes notwithstanding any copyright annotation thereon. Disclaimer: The views and conclusions contained herein are those of the authors and should not be interpreted as necessarily representing the official policies or endorsements, either expressed or implied, of IARPA, DoI/NBC, or the U.S. Government.

## Footnotes

[1] http://www.nist.gov/itl/iad/mig/med13.cfm

[2] http://www.R-project.org/

[3] http://www.cs.cmu.edu/~lujiang/spld

[4] https://code.google.com/p/cuda-convnet/

[5] http://www.robots.ox.ac.uk/~vgg/software/homkermap/

[6] http://cran.r-project.org/web/packages/randomForest/index.html

[7]http://cran.r-project.org/web/packages/ada/index.html

[8]http://cran.r-project.org/web/packages/LiblineaR/index.html

[9]http://cran.r-project.org/web/packages/kernlab/index.html

[10] http://www.di.ens.fr/~laptev/actions/hollywood2/

[11] http://vision.stanford.edu/Datasets/OlympicSports/

[12] http://cran.r-project.org/web/packages/kernlab/index.html