[Reviews · NeurIPS 2014]

Submitted by Assigned_Reviewer_3

This paper proposes an incremental but very sensible and practical modification to 'curriculum learning'. Given a partition of the training examples into classes, they propose an additional regularising term (and an additional parameter) to ensure that the 'easy' examples selected during learning are spread across the classes, and not from one class. The partition into classes can come from a clustering algorithm, or from a priori knowledge.

The idea is straightforward and sensible, and the authors propose an algorithm that looks efficient and correct.

This idea is sufficiently straightforward, plausible and practical that I would expect it to be a natural method to use in practice for curriculum learning (if it has not been used already...)

The authors have conducted some large scale experiments that appear to show that their method has great promise. I found that even with additional explanations of the experiments in the supplementary information, the description of the experiments was still hard to follow, and it was difficult to see exactly what had been done. I feel that, if the paper is accepted, the authors should spend some effort in rewriting the experimental section to make it clear what exactly has been done, and exactly what algorithms have been used and compared: sorry, but I did not find this section clear at all.
Summary: The paper proposes a simple modification to curriculum learning that seems most plausible: they propose adding an additional regularising term to ensure that even a small 'curriculum' (subset of easy examples) is chosen from a range subsets in a (previously defined) partition of the training data. This partition might be defined a priori, or else it could be the output of a clustering algorithm.

There are extensive experiments which appear to show that this method is very promising: I did not find the experimental section to be clearly written.

This is a nice paper that proposes a simple and plausible algorithm that could readily be used in practice. It looks like a useful paper describing a neat incremental improvement to a practical algorithm.

Submitted by Assigned_Reviewer_13

In this paper, the authors introduce a formalism for self-paced learning with diversity. This is a type of “curriculum learning.” Curriculum learning, inspired by the way humans and animals appear to learn, suggests learning training examples in a particular order, for example attempting to fit easy examples prior to hard ones. One challenge is how to decide what constitutes an easy or hard example. Self-paced learning showed that you can use the loss function you are trying to minimize as a measure of difficulty. By weighting the samples you can focus on learning easier ones first. Learning proceeds by minimizing an objective function which has two terms: the weighted loss function over the samples and the negative of the L_1 norm of the weights. A parameter lambda controls the importance of the two terms and is increased while fitting to force the model to incorporate increasingly difficult examples. Because this function is biconvex it is possible to optimize relatively efficiently.

This work extends self-paced learning by incorporating “diversity.” That is, the model should learn not just easy examples, but diverse examples. To achieve this, the authors assume that examples are already grouped in some way (for example, using a clustering algorithm). They propose adding a third term to the objective function, which favors weighting samples from multiple groups (and an additional parameter, gamma, to weight the importance of this term). The objective function is no longer biconvex, however, they show a simple algorithm which they prove is able to find the global minimum of the latent weights in linear time.

The submission then compares the results of objective functions learned using various methods (including self-paced learning) on three datasets: multimedia event detection and two action recognition datasets. Learning with diversity outperformed self-paced learning (and other methods) for all datasets, including apparently fitting in fewer iterations.

This submission builds on previous work to construct a novel approach to curriculum learning. It demonstrate that diversity may be an important property to consider in achieving good fits. It provides a well-constructed rational for the approach, a novel algorithm including proof of optimality and several well-constructed empirical tests. The approach results in significant improvements in several of the datasets considered, which are quite challenging tasks. The authors propose several possible extensions to their approach for further research. Additionally, the use of “diversity” during learning may have some generality, and may be applied in other contexts.

The manuscript is well presented and does an excellent job of conveying the key ideas, while including proofs and additional detail in the supplementary.

One weakness of the manuscript, is that it lacks empirical comparisons with some more known tasks and datasets (for example, MNIST, as in the original SPL work). The datasets chosen here both have obvious diversity (examples from the same category that are very different) and levels of difficulty (examples which are obviously more ambiguous than others). This makes them good choices for testing the ideas presented. However, it would have been helpful to examine well-studied datasets such as MNIST which do not necessarily have the same obvious levels of diversity and difficulty. Does this approach still result in improved fits? It is not obvious to me what the answer would be and it would motivate helpful discussion about the scope of this work.
Summary: A novel contribution to curriculum learning by proposing the idea and methods to utilize diversity during. Well-presented, thought-provoking with compelling empirical results.

Submitted by Assigned_Reviewer_16

An approach to incorporating diversity in self-paced learning is presented, where diversity refers to weight variety over different groups of training examples. By including a regularization term enforcing this notion of diversity, the self-paced learning scheme encourages weighting of examples from different groups to be non-zero.
The proposed approach does appear to be novel and introduces an interesting notion of diversity among groups, however the manner in which groups are generated is an issue requiring more attention. In the experimental section, it appears that all the groups are generated based on spatial distribution. Using groups generated from outside information would be a more interesting application of the approach.
The comparison to random forests and AdaBoost do not appear to be relevant in the experimental section, as these approaches appear to overtrain the detectors, with only 10 positive examples for each class in a 1000 dimensional space. A more interesting comparison would have been with other regularization schemes on linear classifiers such as sparse linear classifiers. Additionally, a comparison to learning on clustered data would be a highly interesting comparison to demonstrate that the gains are not simply due to the clustering preventing overfitting and in fact due to the diversity in self-paced learning.
Summary: The proposed approach is interesting and shows a marginal improvement over tested approaches, however discussion on how groups are generated and additional experimental comparisons (notably a comparison with clustering before applying self-paced learning) need to be included.
Author Feedback
Author rebuttal: We would like to thank the reviewers for their helpful suggestions, and encouraging assessment of our paper.

To R13:

Thanks for precisely introducing the details and illustrating the insight of our work.

On the dataset selection:
Thanks for the helpful suggestion. We selected the experimental datasets based on the consideration that it may be more convincing to show SPLD’s improvements on real-world challenging datasets which naturally tend to have levels of difficulty and evident diversity. Yet, we agree that the comparison on relatively neat data like MNIST is also beneficial for a more complete comparison with SPL. We will add this type of comparisons in our future work.

To R16:

Thanks for the helpful suggestion.

On groups generated from outside information:
Yes, we agree that incorporating the outside “oracle” information seems to be a more sensible choice when such “oracle” or prior is available. In the absence of such information (e.g. in our experimental datasets), however, we have to generate groups by data themselves, which, as shown in our experiments, also proves to be a reasonable solution in practice.

On the comparison to random forests and AdaBoost:
Random forests and Adaboost are two classical training models on sampled data, which may be less relevant yet interesting to compare against since the SPL method as well as our proposed SPLD is also constructed on sampled data. Given that the comparison seems not to compromise the quality of this paper, we include the comparison for “completeness”, and our focus (Para 3 Section 5.1) is still on comparing among BatchTrain, SPL and SPLD. We will clarify this point in the paper.

On the sparse regularization schemes (such as sparse linear classifier):
The purpose of this comparison is to show the advantage of curriculum learning (including SPL and the proposed SPLD) over the batch-mode training (BatchTrain). In our experiments, the BatchTrain method is a typical regularized linear classifier used in related work [8, 23, 20]: linear SVM (hinge loss + l2 regularizer). We agree that it is interesting to compare sparse linear classifier, which sparsely selects features but it still belongs to the BatchTrain family, and empirically may not perform as well as L2 regularizer, as suggested in the related work.

On additional experimental comparison of generated groups:
We highly appreciate this comment. Though we did not put the result in paper, we listed the comparison in Table 4 of the supplementary materials, where we compared the clusters generated from data by two clustering algorithms (K-means and Spectral Clustering) under different number of clusters (32, 64, 128 and 256). As we see, SPLD outperforms SPL across all settings, which suggests that improvement mainly comes from the diversity. We will put Table 4 in the paper and make this point clearer in the discussion.

On discussion of group generation:
We will substantiate our discussion on how groups are generated in the paper.

To R3:

We appreciate the reviewer’s statement that our idea is sufficiently straightforward, plausible and practical.

On rewriting the experimental section:
Sorry for the confusion. We will rewrite the experimental section with effort to make it clearer and more readable. Besides, we will provide instructions for reproducing the experimental results.